# Position: Preregister Experiments with AI Agents

Michelle Vaccaro [1]

## Abstract

The proliferation of large language models (LLMs) and autonomous AI agents has given rise to a rapidly growing methodological paradigm: "in silico" behavioral experiments. Originally conceived as a way to use AI agents as proxies for human participants in studies of cognition, decision-making, and social dynamics, this approach has taken on new significance—as AI agents increasingly negotiate, transact, and make consequential decisions on behalf of people and organizations, understanding their behavior has become a research priority in its own right. While these experiments with AI agents offer unprecedented advantages in terms of scalability, cost efficiency, and experimental control, they also inherit, and in some cases amplify, methodological vulnerabilities that have long plagued human subjects research. To address these issues, this paper argues that preregistration practices—central to improving the credibility of human subjects experiments—should now be extended to experiments with AI agents. We systematically catalog the researcher degrees of freedom that experiments with AI agents introduce—model selection, prompt wording, settings, and outcome-contingent redesign, for example—and show how the low cost of iteration and lack of reporting norms make these choices both easy to exploit and difficult to detect. We propose a preregistration template tailored to experiments with AI agents and call on conferences, journals, and funding agencies to make preregistration standard practice for this emerging research paradigm.

[1]Institute for Data, Systems & Society (IDSS), Massachusetts Institute of Technology, Cambridge, MA, United States. Correspondence to: Michelle Vaccaro <vaccaro@mit.edu>.

*Proceedings of the 43rd International Conference on Machine Learning*, Seoul, South Korea. PMLR 306, 2026. Copyright 2026 by the author(s).

## 1. Introduction

A growing body of machine learning research treats large language models (LLMs) as participants in behavioral experiments and studies their responses to economic games, cognitive tasks, moral dilemmas, and social scenarios to characterize their reasoning, biases, and alignment properties (Horton et al., 2023; Binz & Schulz, 2023; Dominguez-Olmedo et al., 2024; Aher et al., 2023; Scherrer et al., 2024). These experiments with AI agents require a fraction of the time and cost of traditional human subjects research, and can produce response patterns that meaningfully resemble human behavior (Aher et al., 2023; Cao et al., 2025; Sarstedt et al., 2024; Mei et al., 2024). They are also increasingly valuable in their own right as AI agents assume autonomous roles in high-stakes domains—negotiating contracts, managing portfolios, moderating platforms, coordinating infrastructure—and interact with one another in multi-agent systems where their behavior carries real-world consequences (Bianchi et al., 2024; Li et al., 2024).

The social and behavioral sciences have spent much of the past two decades grappling with a replication crisis driven largely by researcher degrees of freedom: flexible, often undisclosed choices about data collection, analysis, and reporting that can inflate false-positive rates even without intentional misconduct (Simmons et al., 2011; Gelman & Loken, 2013; Open Science Collaboration, 2015; Steegen et al., 2016). Experiments with AI agents inherit these vulnerabilities and introduce new ones. Prompt wording, model selection, decoding parameters, retry policies, and response parsing all constitute high-dimensional and high-consequence choice surfaces that researchers can traverse—intentionally or inadvertently—until desired results emerge (Ma et al., 2025; Herrera-Poyatos et al., 2025; Oh et al., 2025; Sclar et al., 2024). The combinatorial explosion of prompt $\times$ model $\times$ temperature $\times$ seed $\times$ parsing configurations creates a vast multiverse of experimental specifications, yet papers typically report only one path through it (Steegen et al., 2016; Gelman & Loken, 2013). Given the relatively low marginal cost of running additional configurations with AI agents, specification search can become routine and largely invisible, blurring the boundary between confirmatory tests and iterative exploration (Simmons et al., 2011; Gelman & Loken, 2013).

This problem is distinct from, though related to, familiar concerns about reproducibility in ML evaluation (Hutson, 2018; Henderson et al., 2018; Gundersen & Kjensmo, 2018). In response to these concerns, the community has developed norms around held-out test sets, fixed training, validation, testing splits, and standardized benchmarks precisely to prevent overfitting to evaluation data (Dwork et al., 2015; Liang et al., 2022; Pineau et al., 2021; Kapoor & Narayanan, 2023). Yet behavioral experiments with AI agents often lack analogous safeguards: there is no "test set" to hold out when the experiment is the evaluation, and no standardized protocol when prompts are bespoke research instruments (Horton et al., 2023; Dominguez-Olmedo et al., 2024; Aher et al., 2023). The result is a methodological gap—one that risks undermining the credibility of an otherwise promising research direction.

To address this issue, we argue that preregistration practices—central to improving the credibility of human subjects experiments—should now be extended to experiments with AI agents. To position preregistration—the practice of publicly committing to hypotheses, methods, and analysis plans before data collection—as standard practice in experiments with AI agents, we structure our paper around the following contributions. First, we review the replication crisis in human subjects research and the role preregistration has played in addressing it (Section §2). Second, we systematically catalog researcher degrees of freedom specific to experiments with AI agents, mapping them onto the $p$-hacking dynamics that motivated preregistration in the behavioral sciences (Section §3). Third, we introduce a preregistration template tailored to experiments with AI agents that addresses these novel degrees of freedom while preserving the exploratory flexibility essential to scientific progress (Section §4). Finally, we offer recommendations for researchers, reviewers, and venues to establish preregistration as standard practice (Section §5).

By learning from the experience of the social and behavioral sciences, the ML community has an opportunity to build credibility into this research paradigm from the outset rather than retrofitting it after a crisis.

## 2. The Replication Crisis in Human Subjects Research

Over the past two decades, psychology, behavioral economics, and adjacent social sciences have confronted a "replication crisis": a growing recognition that many published findings are less reliable than their prominence suggests (Open Science Collaboration, 2015; Camerer et al., 2016; 2018). Early meta-research argued that selective reporting, low statistical power, and strong incentives for novelty can yield a literature with inflated effect sizes and excess false positives (Ioannidis, 2005; Button et al., 2013). As awareness grew, attention shifted from outright fraud to the subtler mechanics of researcher degrees of freedom: flexible decisions about exclusion rules, stopping, outcome definitions, and model specifications that can "discover" significance even when evidence is weak (Simmons et al., 2011), especially in the "garden of forking paths" where many defensible analytic routes exist (Gelman & Loken, 2013). Survey evidence reinforced that such questionable research practices were not rare edge cases but part of routine scientific workflow (John et al., 2012). Large-scale replication efforts subsequently quantified the problem: the Open Science Collaboration's 2015 project found that while 97% of original studies reported statistically significant results, only 36% of replications were significant, with replication effects about half the magnitude of originals (Open Science Collaboration, 2015). Parallel efforts in experimental and behavioral economics echoed these concerns, finding materially lower replication rates and attenuated effect sizes even under high-powered designs (Camerer et al., 2016; 2018).

In response to the replication crisis, preregistration emerged as one of the most widely endorsed methodological reforms. At its core, preregistration requires researchers to publicly commit to hypotheses, methods, and analysis plans before data collection, so it constrains the flexibility that otherwise enables $p$-hacking, HARKing ("hypothesizing after results are known"), and selective outcome reporting (Simmons et al., 2011; Nosek et al., 2018). This process also creates a clear demarcation between confirmatory research, which tests hypotheses specified in advance, and exploratory research, which generates hypotheses from observed patterns—a distinction that is epistemically crucial but easily blurred in standard publication practice (Wagenmakers et al., 2012). Importantly, the idea of preregistration was not new to science: clinical trials had long required registration, and ClinicalTrials.gov, launched in 2000, now contains over half a million entries (National Institutes of Health (NIH), 2025; National Library of Medicine, 2025). While the social and behavioral sciences adopted preregistration later, it has become increasingly mainstream—normalized by low-friction platforms such as OSF Registries (now hosting 100,000+ public registrations) and streamlined services like AsPredicted, and reinforced by growing editorial expectations as more journals encourage or prioritize preregistered studies and expand pathways like Registered Reports (Pfeiffer & Call, 2022; Credibility Lab, 2026; Center for Open Science, 2024; Lin et al., 2024). Early evidence suggests these reforms are working: preregistered studies tend to report smaller effect sizes and higher rates of null results than their non-preregistered counterparts, consistent with reduced publication bias and analytic flexibility (Schäfer & Schwarz, 2019; Scheel et al., 2021; van den Akker et al., 2024).

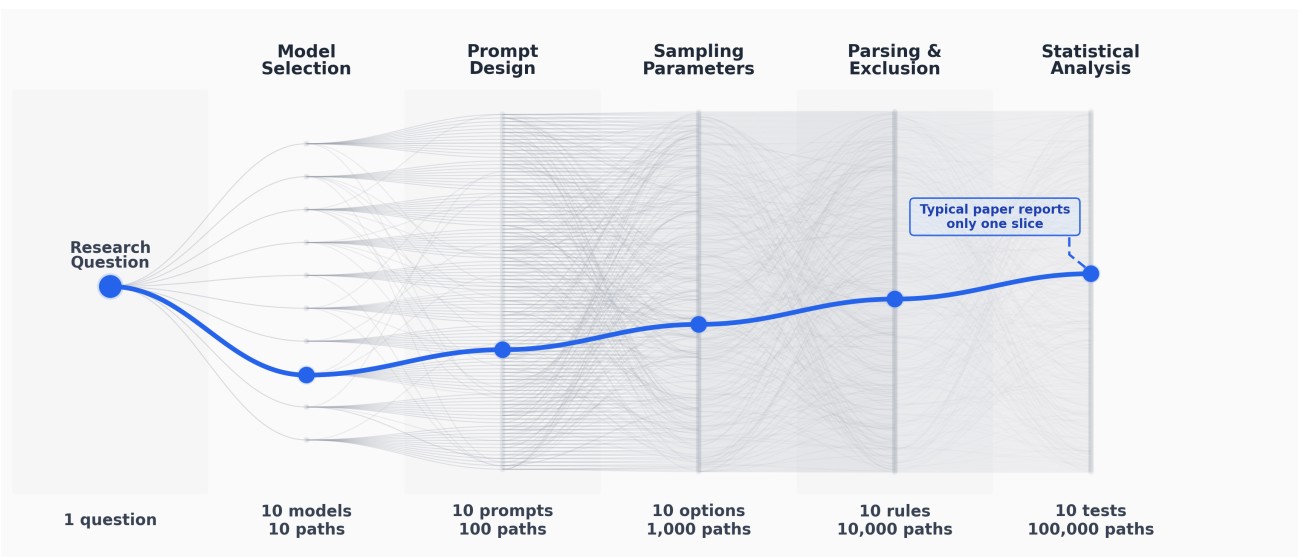

*Figure 1.* **Garden of forking paths (Gelman & Loken, 2013) in experiments with AI agents.** In experiments with AI agents, a single research question branches into a combinatorial space of experimental configurations as researchers make choices about model specifications, prompt design, sampling parameters, parsing and exclusion rules, and statistical analysis procedures. A preregistered confirmatory study follows a single precommitted path or set of paths (blue slice). In contrast, routine iteration enables specification search (gray lines), where many models, prompts, parameters, rules, and statistical tests are explored but only a single configuration or small subset of configurations are typically reported.

## 3. Researcher Degrees of Freedom in Experiments with AI Agents

Our taxonomy (Table 1) makes explicit that experiments with AI agents inherit the same core problem that motivated the "researcher degrees of freedom" critique in human subjects research: many reasonable-seeming choices—often made implicitly—can be flexibly tuned until a desired pattern appears. In psychology, undisclosed flexibility in data collection, analysis, and reporting can dramatically inflate false-positive rates (Simmons et al., 2011). Even when researchers do not consciously "fish," the garden of forking paths problem shows that a large space of defensible analytic choices can create a multiple-comparisons problem in practice (Gelman & Loken, 2013). Table 1 translates these concerns into the AI setting by organizing degrees of freedom across the full experimental pipeline—from model selection and prompt engineering through sampling parameters, experimental design, response processing, analysis, and reporting.

Many of these AI-specific degrees of freedom are easy to vary, consequential for results, and lack principled defaults. Small prompt perturbations can produce large downstream differences in outputs, effectively turning "prompt wording" into a high-dimensional treatment manipulation (Salinas & Morstatter, 2024; Zhao et al., 2021; Liu et al., 2023). Likewise, stochastic decoding and randomness control (temperature, top-$p$, seeds) can meaningfully alter response content and refusal behavior, creating a danger of re-running,

filtering, or "stabilizing" outputs until they align with expectations (Larsen, 2025). Relatedly, inference budget is not a neutral implementation detail: increasing tokens, turns, or tool calls can change not only mean performance but also which strategies the agent adopts, making compute allocation itself a potential (and often unacknowledged) moderator (Snell et al., 2024; Wei et al., 2022; Wang et al., 2023). Unlike traditional experimental stimuli, which are typically fixed before data collection begins, prompts are often iteratively refined in response to model outputs—a process that blurs the boundary between pilot testing and confirmatory experimentation (Zamfirescu-Pereira et al., 2023; Nosek et al., 2018). The same ambiguity applies to model selection: with dozens of available models differing in capability, alignment tuning, and behavioral idiosyncrasies, the choice of which model to report can itself become an outcome-contingent decision (Pineau et al., 2021; Raff, 2019).

Importantly, these degrees of freedom interact multiplicatively—prompt × model × decoding × retries × parsing × metric—yielding a combinatorial multiverse of plausible experimental specifications (see Figure 1). These problems echo those in psychology, which has responded with calls to assess robustness across alternative analytic pipelines rather than presenting a single privileged path (Steegen et al., 2016). Yet the multiverse in AI experiments is vastly larger: a study might reasonably vary prompt wording across a dozen framings, test three model families each with multiple versions, explore several temperature settings, implement different retry and refusal-

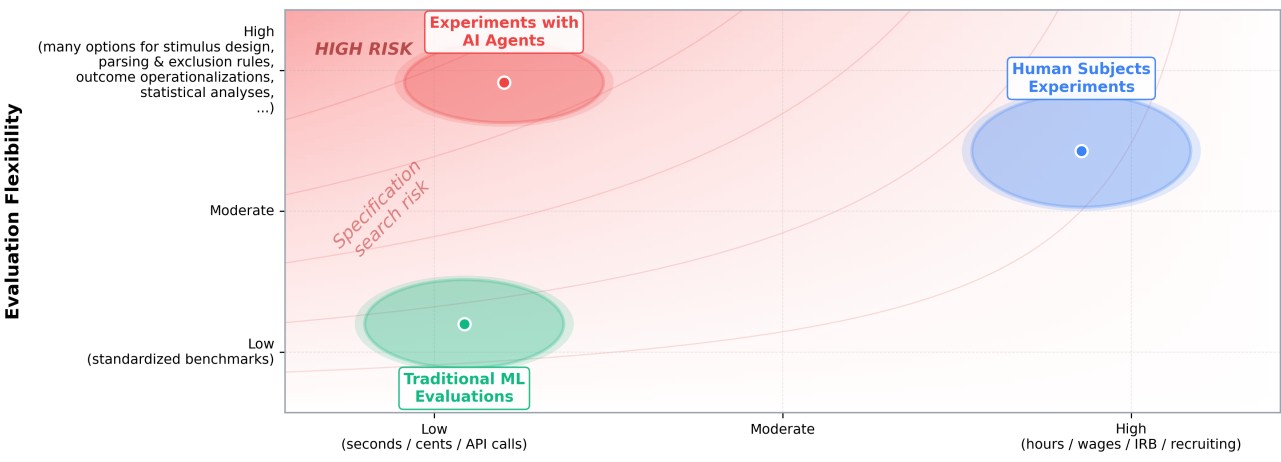

*Figure 2.* **Cost–flexibility tradeoff across research paradigms.** The risk of specification search depends jointly on two factors: the marginal cost of running additional experimental configurations ($x$-axis) and the degree of evaluation flexibility in the experiment ($y$-axis). Human subjects behavioral experiments (blue) involve substantial evaluation flexibility but high marginal costs—participant recruitment, compensation, IRB oversight, and data collection time—that impose natural friction on exhaustive exploration. Traditional ML benchmarking (green) is inexpensive to rerun, but fixed test sets and standardized metrics constrain evaluation flexibility, limiting the returns to extensive specification search. Experiments with AI agents (red) occupy a distinctive region: the marginal costs of additional trials are low (seconds and cents per API call) while evaluation flexibility remains high, encompassing stimulus design, model and version selection, decoding parameters, retry policies, parsing and exclusion rules, outcome operationalizations, and statistical tests, to name a few.

handling policies, and choose among multiple outcome operationalizations. The full specification space can easily encompass thousands of defensible configurations, each capable of producing different results. Reporting only the final configuration—without acknowledging the paths not taken—obscures the true uncertainty surrounding the findings (Steegen et al., 2016; Simonsohn et al., 2020; Botvinik-Nezer et al., 2020).

What makes this problem especially acute for AI agent experiments is the conjunction of high flexibility and low marginal cost (see Figure 2). Human subjects research involves substantial analytic flexibility, but the expense of participant recruitment, compensation, IRB oversight, and data collection imposes natural friction on exhaustive exploration. Traditional ML benchmarking operates at low cost, but standardized test sets and metrics constrain the specification space, and the field has increasingly emphasized reporting standards and reproducibility checklists to mitigate selective disclosure (Pineau et al., 2021; Mitchell et al., 2019). Experiments with AI agents occupy a distinctive and concerning region: running an additional experimental configuration only requires seconds and pennies, but the space of defensible specifications remains quite vast. This asymmetry creates strong incentives—and unprecedented opportunities—for specification searches that would be prohibitively expensive with human participants.

Compounding the problem, many of these specification choices leave no visible trace in published work. A paper might report results from GPT-4 without mentioning that Claude and Gemini were also tested; it might present a particular prompt without acknowledging dozens of prior iterations; it might describe a parsing rule without noting that alternatives were tried and abandoned. Unlike preregistered human subjects experiments, where deviations from protocol are increasingly expected to be disclosed, norms for transparency in AI experiments remain underdeveloped (Gundersen & Kjensmo, 2018). The result is a literature in which the reported findings may reflect not robust phenomena but rather the outcome of an undisclosed optimization process over the specification space (Gelman & Loken, 2013; Simmons et al., 2011).

To make the potential for specification-driven variability concrete, we conducted a simulation examining anchoring effects in LLMs across 2,430 experimental specifications, varying model family, system prompt, anchor distance, delivery method, question content, and outlier handling (see Section A.1 and Figure 3). The resulting specification curve shows anchoring indices ranging from strongly negative to strongly positive—a researcher could, depending on which path through the specification space they report, conclude that LLMs exhibit robust human-like anchoring, no anchoring whatsoever, or even reverse anchoring. A researcher could even report something like: "Across three models

from different families and architectures, we find consistent evidence that LLMs exhibit human-like anchoring biases"—a claim that sounds like a compelling robustness check but was selected from a much larger space of possible analyses, most of which would tell a different story. The result is an illusion of robustness: a finding that appears to generalize across models and conditions, but whose apparent generality reflects the researcher's (conscious or unconscious) navigation of the specification space rather than a stable property of the phenomenon under study.

# 4. Preregistration for Experiments with AI Agents

To address these challenges, we propose a preregistration template specifically designed for experiments with AI agents (see Supplementary Materials). The template extends the logic of traditional preregistration—constraining flexibility by requiring advance commitment to hypotheses, methods, and analyses—while adding structured fields for the novel degrees of freedom that characterize AI agent research. In this section, we describe the template's core components and explain how each targets specific threats to inference identified in our taxonomy (Table 1).

## 4.1. Specifying models, parameters, and prompts

This template requires complete specification of the computational environment before data collection begins. Researchers specify exact model identifiers and version checkpoints (e.g., `gpt-4-0125-preview` rather than simply "GPT-4"), generation parameters (temperature, top-$p$, top-$k$, random seeds), and inference-budget constraints (maximum tokens, timeout thresholds, retry limits). For API-accessed models, it is important to include the API version and access date, since providers may update model behavior over time in ways that complicate reproducibility even when names or endpoints appear stable (Chen et al., 2024). Likewise, for open-weight models, it is important to specify the exact checkpoint hash and any quantization scheme.

The template also requires specifying the complete prompt text provided to the agents, including system messages, user instructions, and any few-shot examples. The prompt specifications should be verbatim—not paraphrased summaries—since even subtle formatting or wording changes can produce meaningful behavioral shifts (Sclar et al., 2024; Oh et al., 2025). For studies using retrieval-augmented generation or dynamic context, researchers should also specify the retrieval mechanism, document corpus, and any filtering or ranking procedures applied to retrieved content (Lewis et al., 2020). The goal is to ensure that another researcher could, in principle, reproduce the experimental conditions—a standard that much current AI research still struggles to meet (Gundersen & Kjensmo, 2018; Pineau et al., 2021; Hutson,

2018; Henderson et al., 2018).

## 4.2. Documenting pilot testing and development history

Importantly, the preregistration requires disclosure of pilot-testing history: how many prompt iterations were explored, which models were tested during development, and whether any preliminary results informed the final hypotheses. This provision directly targets the concern that the low marginal cost of experiments with AI agents invites extensive specification search that remains invisible in final manuscripts. A researcher who tested fifteen prompt variants before settling on the registered version has not necessarily engaged in misconduct—prompt development is often a legitimate part of experimental design—but this history is epistemically relevant. The registered prompt may have been selected precisely because it produced the expected pattern, narrowing the effective hypothesis space in ways that inflate false-positive risk (Simmons et al., 2011; Gelman & Loken, 2013).

## 4.3. Operationalizing the confirmatory–exploratory distinction

Additionally, the template operationalizes the distinction between confirmatory and exploratory research in ways suited to the AI paradigm. Researchers declare which analyses are hypothesis-testing (confirmatory) versus hypothesis-generating (exploratory), and commit to reporting each as such. This distinction matters because the norms governing inference differ: confirmatory analyses warrant conventional statistical interpretation, while exploratory findings should be framed as hypothesis-generating and evaluated under different evidentiary standards (Wagenmakers et al., 2012).

For confirmatory analyses, the template requires pre-specification of the primary outcome variable(s), the statistical test(s) to be applied, and the decision rule for interpretation (e.g., significance threshold, minimum effect size). For studies using LLM-based evaluation (e.g., using GPT-4 to score open-ended responses), researchers also commit to the evaluation prompt, specify the evaluator model version, and describe procedures for handling evaluator disagreement or refusal. This is particularly important given evidence that LLM-based judges can exhibit systematic biases, positional effects, and version sensitivity (Zheng et al., 2023). Furthermore, the template requires pre-specification of exclusion criteria for malformed, refused, or filtered responses, since refusal handling (exclude, impute, retry, or code as a category) can substantially affect results and is otherwise easy to "optimize" post hoc (Simmons et al., 2011; Gelman & Loken, 2013).

### 4.4. Addressing robustness and generalization

The template also includes dedicated sections for studies examining robustness or generalization. The core principle is straightforward: whatever space of specifications a researcher plans to explore must be defined in advance, and results from the full space must be reported. If researchers intend to test whether findings replicate across prompt variants, all variants must be listed beforehand and reported afterwards—including those that yielded null or contrary findings (Steegen et al., 2016). If testing generalization across models, all models must be specified upfront, and results from every model must be reported—not just those yielding preferred outcomes. The same logic applies to parameter-sensitivity analyses: the parameter ranges to be explored and the method for aggregating results across those ranges must be committed to in advance. Without these provisions, "robustness checks" can easily become selectively reported post-hoc exercises that lend false credibility to preferred conclusions rather than genuinely test them. A study that pre-commits to testing three models and reports results from all three provides far stronger evidence than one that tested six but reports only the three that "worked."

### 4.5. Handling staged and adaptive designs

We recognize that some legitimate research designs require sequential decision-making—for instance, using pilot data to calibrate stimulus difficulty or determine appropriate sample sizes. The template accommodates such designs through a staged preregistration option, borrowing from Registered Report frameworks (Chambers, 2013). Researchers may preregister an initial design with explicit decision rules for subsequent stages: "If pilot data indicate floor effects (accuracy $< 20\%$), we will increase context length by 500 tokens; if ceiling effects (accuracy $> 90\%$), we will use more difficult items from the backup set." The key constraint is that these decision rules must be specified in advance, not invented post hoc.

For genuinely exploratory work—which remains valuable and necessary—the template does not demand artificial constraints. Instead, it asks researchers to clearly label such work as exploratory and to separate it from confirmatory claims. The goal is not to eliminate flexibility, but to make flexibility visible, allowing appropriate calibration of confidence in reported findings (Simmons et al., 2011; Gelman & Loken, 2013).

### 4.6. Transparency and reproducibility commitments

Finally, the template embeds transparency commitments that facilitate cumulative science. Researchers must specify whether raw model outputs, processed data, and analysis code will be shared, and where. For studies using proprietary models or APIs, we encourage archiving complete input–output logs, since model behavior may change over time even for fixed version identifiers (Chen et al., 2024). The template also includes fields for linking to code repositories, data archives, and supplementary materials, creating a complete audit trail from preregistration through final manuscript.

Furthermore, an attestation section requires confirmation that data collection has not yet begun—addressing concerns that the ease of re-running AI experiments makes post-hoc registration tempting. The attestation does not guarantee compliance, but it raises the reputational stakes for violating the commitment and signals to readers that the researchers intended a genuinely confirmatory test.

### 4.7. Balancing rigor and practicality

We acknowledge that comprehensive preregistration imposes costs. Specifying every detail in advance requires more careful upfront planning, and strict adherence may occasionally force researchers to report analyses they later recognize as suboptimal (Nosek et al., 2018). These costs are real but, we argue, worth bearing. The alternative—a literature of unreported flexibility and unknowable false-positive rates—imposes larger costs on the field as a whole (Simmons et al., 2011). Moreover, preregistration need not be inflexible: deviations from the registered plan are acceptable when clearly disclosed and justified, and exploratory analyses remain valuable when labeled as such (Chambers, 2013; Wagenmakers et al., 2012).

By making the full specification explicit and time-stamped, the template enables reviewers and readers to evaluate not just whether reported results are statistically significant, but whether they emerge from a constrained hypothesis test or a broader search through the garden of forking paths (Gelman & Loken, 2013). As AI agents become increasingly central to the behavioral sciences—both as objects of study and as tools for studying humans—establishing rigorous methodological standards is essential for building a cumulative, trustworthy science of AI behavior.

## 5. Call to Action: Recommendations for Stakeholders

### 5.1. For Researchers

Researchers conducting experiments with AI agents should treat preregistration not as bureaucratic overhead but as a tool for credible inference. We recommend registering studies on established platforms such as OSF Registries before any data collection begins, using templates adapted for AI research that capture model specifications, exact prompts, and generation parameters, like the proposed template in

this paper. Pilot testing to refine prompts and select models is both legitimate and encouraged, but the boundary between piloting and confirmatory testing should be explicit in the preregistration. When exploratory analyses reveal unexpected patterns, researchers should report them transparently as exploratory and, where warranted, conduct separate preregistered follow-up studies to confirm. We also encourage researchers to adopt multiverse or specification-curve approaches when feasible, reporting results across plausible combinations of models, prompts, and parameters rather than presenting a single privileged configuration. Finally, sharing complete materials—including raw model outputs, full prompt text, and analysis code—should become default practice, enabling independent verification and cumulative refinement of methods.

### 5.2. For Conferences and Journals

Conferences and journals serve as gatekeepers of scientific quality and are well positioned to normalize preregistration in this emerging field. We recommend that venues publishing experiments with AI agents explicitly encourage or require preregistration, following the model of journals that have adopted Registered Reports or badge systems for open practices. At minimum, submission guidelines should ask authors to disclose whether studies were preregistered and, if not, to justify why. Reviewer guidelines should also be updated to prompt evaluation of researcher degrees of freedom specific to AI experiments: Were model and prompt specifications shared in advance? How many configurations were tested? Are robustness checks genuine or post-hoc? Venues might also consider dedicated tracks or submission categories for preregistered studies, providing incentives for researchers to adopt these practices. For papers reporting non-preregistered work, reviewers should expect comprehensive robustness analyses and full transparency about the specification search process, with results appropriately framed as exploratory or preliminary.

### 5.3. For Funding Agencies

Funding agencies shape research norms through the priorities they set and the requirements they impose. Agencies supporting research involving AI agents as experimental participants should consider requiring preregistration as a condition of funding, paralleling existing mandates for clinical trial registration in biomedical research (National Institutes of Health, 2016). Grant application templates could also include sections asking investigators to describe their preregistration plans, specify which platforms they will use, and explain how they will handle the unique degrees of freedom in AI experimentation. Finally, funding agencies should recognize that preregistration represents a cultural shift that requires training and support. Investing in workshops, tutorials, and educational materials—particularly for

researchers from machine learning communities less familiar with preregistration norms—could accelerate adoption and improve the quality of funded research.

## 6. Alternative Views

**Alternative View 1: The low cost of AI experiments makes preregistration pointless.** Horton et al. (2023) **argue that preregistration works for traditional experiments because high fixed costs create natural friction against specification search, but "when it costs \$1 and 30 seconds to run an experiment, it is hard to see what the benefit would be."** We argue that low marginal cost is precisely what makes preregistration more necessary for experiments with AI agents, not less. When running another configuration is trivially cheap, researchers may, intentionally or unintentionally, search specifications until they find one that "works"—and this flexibility is invisible in final manuscripts. Preregistration provides the friction that cost no longer does. Notably, preregistration for meta-analyses faces the same objection—analyzing existing literature is essentially costless and endlessly re-analyzable—yet PROSPERO and similar registries are now widely endorsed precisely because the flexibility inherent in inclusion criteria, outcome selection, and analytic choices demanded constraint (Stewart et al., 2012). The same logic applies to AI experiments.

**Alternative View 2: Preregistration can be manipulated. Nothing stops researchers from exploring widely, then preregistering only the final cleaned-up pipeline that worked—creating an illusion of confirmatory rigor over what was actually extensive specification search.** Our proposed template addresses this vulnerability by requiring researchers to disclose their pilot testing history: how many prompt iterations were explored, which models were tested, and whether preliminary results informed final hypotheses. Dishonest researchers can always lie, but preregistration creates an auditable record that makes deception riskier and easier to detect, especially when combined with norms of sharing raw outputs and analysis code.

**Alternative View 3: Preregistration privileges null hypothesis testing. AI behavioral research would benefit more from estimation, prediction, or machine learning approaches where the confirmatory/exploratory distinction carries less epistemic weight.** Preregistration doesn't have to privilege null-hypothesis significance testing—it can preregister estimands, loss functions, evaluation metrics, and decision thresholds for prediction or ML-style models just as well as $p$-values. Even in estimation or forecasting, the core problem remains: flexible choices about prompts, models, features, hyperparameters, and evaluation can be tuned until performance "looks good," so the confirmatory/exploratory boundary still matters for interpreting

generalization claims. A preregistered plan can explicitly focus on estimation (e.g., effect sizes with intervals), prediction (held-out protocols, predeclared splits), or robustness (specification curves) while still forcing transparency about what was locked in advance versus discovered post hoc. In short, preregistration is a commitment device for research degrees of freedom, not a commitment to NHST.

**Alternative View 4: Mandatory preregistration could kill serendipity. Some of the field's most important findings have emerged from unexpected model behaviors discovered during unplanned exploration—precisely the kind of discovery that rigid precommitment might discourage.** This concern is legitimate: science benefits from surprise, and we should be wary of bureaucratic requirements that penalize intellectual flexibility. But preregistration does not prohibit exploration; it requires that exploration be labeled as such. Serendipitous discoveries remain fully reportable—researchers simply must distinguish unexpected findings from prespecified hypothesis tests. This transparency actually protects exploratory findings: a surprising result flagged as exploratory and subsequently confirmed in a preregistered follow-up carries far more credibility than one ambiguously presented as predicted all along.

**Alternative View 5: Any single preregistered specification is arbitrary. The combinatorial explosion of prompt × model × parameter configurations means a preregistered design is just one point in a vast multiverse, which undermines claims about generalizability.** This is a real limitation: no single study can establish that findings hold across the full space of reasonable specifications. But preregistration makes clear which slice was tested confirmatorily and prevents quietly searching the multiverse until one slice works. Moreover, researchers can preregister the multiverse itself: a planned set of models, prompt variants, and parameter ranges plus an aggregation rule (specification curve, hierarchical pooling, worst-case summary), evaluating generalizability across a defined space rather than asserting it from a single configuration.

**Alternative View 6: The ML community's existing norm of open-sourcing code is the right credibility mechanism for computational research, because unlike human subjects experiments, AI experiments can be cheaply rerun and independently verified.** We agree that open-sourcing addresses reproducibility—whether someone else can run the same pipeline and get the same numbers—and our template includes it as a core commitment. But open-sourcing the final experiment makes only that single path reproducible; it does not reveal the garden of forking paths that preceded it—the prompt variants, models, temperature settings, and parsing rules explored before settling on the configuration that worked (Gelman & Loken, 2013). This is the distinction between reproducibility (*can I get the same result from the same pipeline?*) and credibility (*should I believe this result reflects a robust phenomenon rather than a specification optimized to produce it?*). Open-sourcing also places the burden of detection on the wrong parties: reviewers under tight deadlines and researchers facing compounding API costs are unlikely to systematically test alternative specifications, meaning the burden often goes undischarged entirely. Preregistration inverts this burden by asking the original researchers—who possess full knowledge of the experimental history—to make their specification choices transparent from the outset.

## 7. Conclusion

AI agents now serve as participants in studies of cognition, decision-making, and social behavior—research that informs not only our understanding of these systems but also the policies governing their deployment in high-stakes domains. Yet this emerging paradigm has developed largely without the procedural safeguards that the social and behavioral sciences spent two decades building in response to their own replication crisis. The researcher degrees of freedom we have catalogued—spanning model selection, prompt engineering, sampling parameters, and reporting practices—create a vast space of experimental specifications that can be traversed, intentionally or inadvertently, until desired results emerge. Without preregistration, the field risks accumulating a literature of findings that appear robust but reflect unacknowledged specification search rather than genuine regularities in AI behavior.

The methodological vulnerabilities of experiments with AI agents are neither minor nor speculative. The taxonomy we present points to a systematic correspondence between the researcher degrees of freedom that inflated false-positive rates in human subjects research—optional stopping, outcome switching, selective reporting—and their analogues in AI experimentation. In some respects, the AI setting actually amplifies these risks: the combinatorial explosion of prompt × model × temperature × parsing configurations creates a multiverse of experimental specifications far larger than what researchers typically navigate in human studies, while the near-zero marginal cost of running additional configurations removes the natural friction that once constrained specification search. The preregistration template we propose directly targets these vulnerabilities, requiring researchers to lock model identifiers and generation parameters, commit to exact prompt text, disclose pilot testing history, and explicitly demarcate confirmatory from exploratory analyses. These provisions do not prohibit exploration—they simply make the research process more transparent, so findings can be evaluated in light of the full specification search that produced them.

The machine learning community has an opportunity that

the social sciences did not: to institutionalize credibility-enhancing practices before a crisis forces retroactive reform. Conferences, journals, and funding agencies are well positioned to normalize preregistration by updating submission guidelines, reviewer instructions, and grant requirements. Researchers can also adopt these practices voluntarily, treating preregistration as a tool for producing more rigorous and impactful work. The goal is not to constrain scientific creativity but to make the research process legible—so that when a study reports that an AI agent exhibits a particular bias, preference, or reasoning pattern, readers can evaluate whether that finding emerged from a principled test or from an undisclosed journey through the garden of forking paths. Building trust in this research paradigm now will pay dividends as AI agents assume ever more consequential roles in society.

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

# A. Appendix

*Table 1.* Taxonomy of Researcher Degrees of Freedom in Experiments with AI Agents

| Category | Degree of Freedom | Issue in Experiments with AI Agents |
|---|---|---|
| Agent Selection | Model family | Selecting LLM providers based on preliminary results |
| | Model version | Switching between model versions after observing initial results |
| | Model parameters | Tuning temperature, top-p, top-k, or seed values to obtain desired responses |
| | Inference budget | Giving some conditions more "thinking room" (max tokens, max turns, timeouts) to get hypothesized performance |
| | Fine-tuned vs. base models | Choosing between fine-tuned and base models based on which yields expected results |
| Agent Scaffolding and Environment | Memory settings | Enabling or disabling conversation memory, adjusting context window usage, or clearing state between trials based on which configuration yields favorable results |
| | Toolchain and sandbox configuration | Selecting which tools, plugins, or execution environments to provide the agent until expected results emerge |
| | System message policies | Placing instructions strategically in system vs. user messages to elicit anticipated behaviors |
| | Multi-agent configuration | Selecting number of agents, roles, or interaction topology to achieve desired outcomes |
| | Concurrency, rate limits, and batching | Varying request timing, batch sizes, or parallelization in ways that drive outputs to confirm hypotheses |
| Prompt Engineering | Prompt wording | Iterating on phrasing, word choice, or framing until expected results emerge |
| | Few-shot example selection | Curating in-context examples that prime the model toward hypothesized responses |
| | Context length | Varying the amount of context provided to achieve desired behaviors |
| | RAG content | Selectively including or excluding retrieved documents that support hypotheses |
| | Instruction ordering | Arranging prompt sections to exploit primacy or recency effects |
| Experimental Design | Task selection | Choosing tasks where the model performs as expected |
| | Stimulus choice | Selecting which vignettes, scenarios, or items to include post hoc |
| | Number of samples per condition | Running variable numbers of completions and selectively reporting |
| Response Processing | Parsing and coding | Adjusting parsing rules or coding schemes after observing which interpretations support hypotheses |
| | Handling refusals/non-responses | Choosing whether to retry, exclude, or impute refusals based on which approach favors expected results |
| | Response validation criteria | Setting or adjusting validity thresholds after observing which criteria yield expected outcomes |
| | Retry policies | Selectively re-querying failed responses or modifying retry logic based on whether retries improve target metrics |
| Analysis Pipeline | Outcome variable operationalization | Defining dependent variables (e.g., sentiment scores, categorical codings) post hoc |
| | Aggregation method | Choosing how to aggregate across runs, agents, or items post-hoc |
| | Statistical test selection | Selecting among tests (e.g., parametric vs. non-parametric) based on results |
| | Covariate adjustment | Including or excluding model-level or prompt-level covariates based on results |
| | Multiple comparisons | Testing many metrics (accuracy, win-rate, rubric score, Elo, pass@k) and reporting the best |
| | Outlier and exclusion criteria | Removing runs, agents, or responses based on post hoc criteria |
| Reporting | Selective reporting | Reporting only models, prompts, or conditions that confirm hypotheses |
| | HARKing | Presenting exploratory prompt iterations or model comparisons as pre-planned analyses |
| | Undisclosed pilot studies | Conducting unreported preliminary experiments to refine design |

## A.1. Illustrative Empirical Demonstration

To illustrate how specification-driven variability operates in practice, we conducted a simulation examining whether LLMs exhibit anchoring effects—the well-documented tendency in human judgment for initial numeric values to bias subsequent estimates (Tversky & Kahneman, 1974; Jacowitz & Kahneman, 1995). Anchoring is a natural candidate for this demonstration as it is one of the most robust and widely replicated findings in human decision-making research (Furnham & Boo, 2011; Röseler et al., 2022).

To demonstrate how the researcher degrees of freedom involved in testing for anchoring in LLMs can generate divergent conclusions, we constructed a full-factorial design varying specifications across the experimental pipeline. We crossed nine models from three families (Gemini, OpenAI, Llama), three system prompt framings (human-like persona, incentivized, none), three anchor distances (high, medium, low), three delivery methods (comparative, embedded, incidental), five estimation questions, and two outlier-handling rules (include, exclude), yielding 2,430 unique experimental specifications. For each specification, we computed an anchoring index, where values near zero indicate no anchoring effect and positive values indicate that estimates were pulled toward the anchor (Jacowitz & Kahneman, 1995). Each specification represents

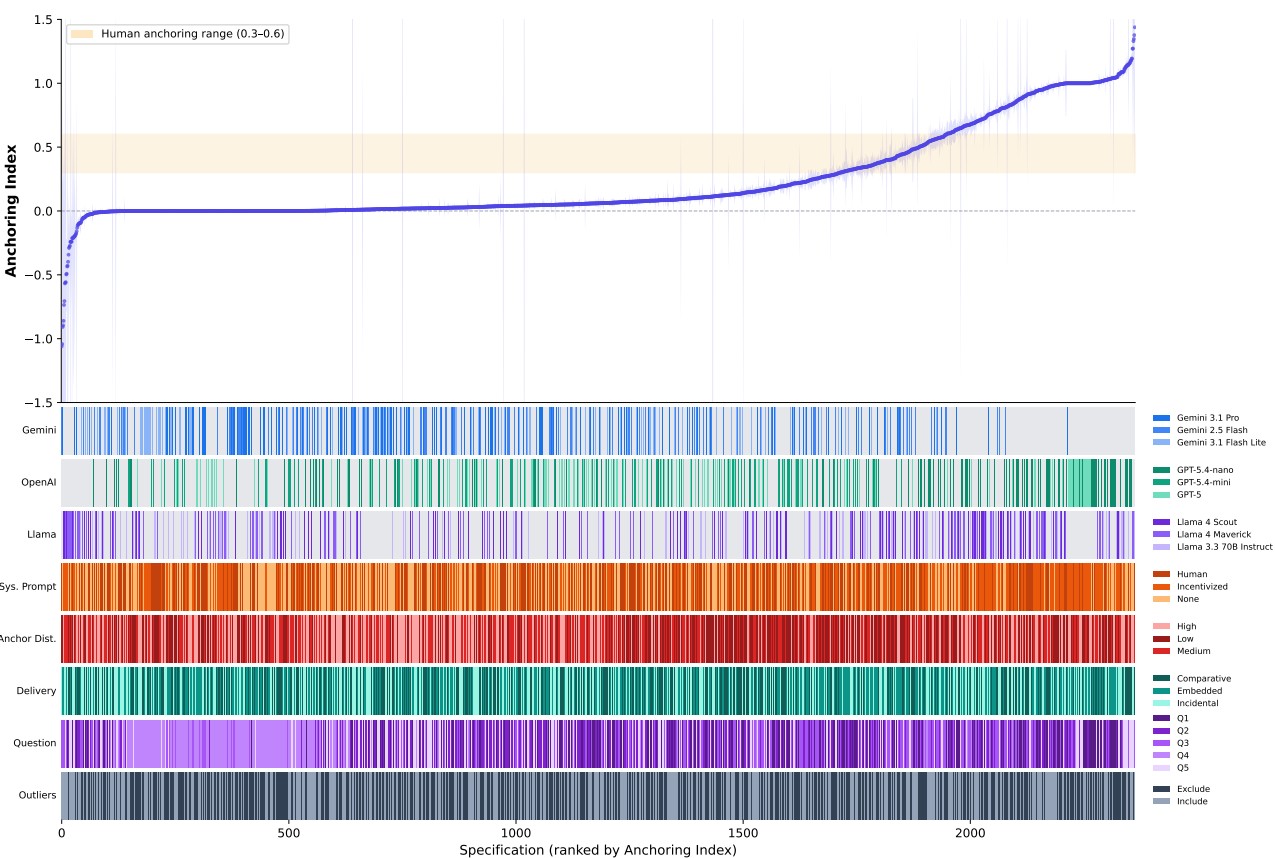

*Figure 3.* **Specification curve for anchoring effects across 2,430 experimental configurations.** Each point represents the anchoring index from a unique combination of model family (9 models across Gemini, OpenAI, and Llama families), system prompt (human-like, incentivized, none), anchor distance (high, medium, low), delivery method (comparative, embedded, incidental), question content (Q1–Q5), and outlier handling (include, exclude). The shaded band indicates the typical range of human anchoring effects (0.3–0.6) (Tversky & Kahneman, 1974; Jacowitz & Kahneman, 1995; Furnham & Boo, 2011; Röseler et al., 2022). Results span from reverse anchoring to effects well above the human range, illustrating how a researcher could reach dramatically different conclusions depending on which path through the specification space they report.

a defensible experimental configuration—a path through the garden of forking paths that a researcher might reasonably choose and report (Gelman & Loken, 2013).

The resulting specification curve (Figure 3) reveals substantial variability across configurations: anchoring indices range from negative values (suggesting reverse anchoring) through zero (no effect) to values exceeding 1.0 (stronger than effects typically observed in human studies) (Jacowitz & Kahneman, 1995; Furnham & Boo, 2011; Röseler et al., 2022). Importantly, this variability means that a researcher's conclusions depend heavily on which slice of the specification space they report. A researcher selecting particular models, prompt framings, and questions could report findings consistent with robust human-like anchoring in LLMs; a different but defensible selection could support the conclusion that LLMs are not susceptible to anchoring at all. We emphasize that this demonstration is illustrative, not a comprehensive study of anchoring in LLMs. We do not claim that anchoring effects in LLMs are or are not real—indeed, the point is that the answer can depend substantially on specification choices. In other words, the space of defensible experimental configurations is large enough, and the resulting variability consequential enough, that a researcher operating without preregistration could—consciously or not—navigate to a preferred conclusion. Full code, data, and outputs are available at https://osf.io/5rc2h/overview?view_only=53e0a1748b4943e4989a2203a1ba9c81.

