# Preregistration Template for Experiments with AI Agents

*Version 1.0*

## Purpose

This template is designed to facilitate preregistration of experiments that use AI agents (e.g., large language models) as participants. It addresses the unique researcher degrees of freedom in AI experimentation: model selection, prompt engineering, sampling parameters, response processing, and analysis choices. Complete this template before data collection begins and submit to a public registry (e.g., OSF Registries).

## 1. Study Information

**Study Title:** *[Enter descriptive title]*

**Authors:** *[List all authors and affiliations]*

**Date of Preregistration:** *[YYYY-MM-DD]*

**Research Questions:** *[State primary research question(s)]*

## 2. Model Specifications

**Model identifier(s):** *[e.g., gpt-4-0125-preview, claude-3-opus-20240229, llama-3-70b-instruct]*

**Model version/checkpoint:** *[Exact version string, checkpoint hash, or API version date]*

**Access method:** *[API endpoint / Local deployment / Cloud service]*

**Access date:** *[Date of API access or model download: YYYY-MM-DD]*

**Generation Parameters:**

| Parameter | Value | Justification |
|---|---|---|
| Temperature | *[value]* | *[optional]* |
| Top-p (nucleus sampling) | *[value]* | *[optional]* |
| Top-k | *[value]* | *[optional]* |
| Random seed(s) | *[value]* | *[optional]* |
| Max tokens (output) | *[value]* | *[optional]* |
| Max tokens (context) | *[value]* | *[optional]* |
| Frequency penalty | *[value]* | *[optional]* |
| Presence penalty | *[value]* | *[optional]* |

| Stop sequences | [value] | [optional] |
|---|---|---|
| Timeout threshold | [value] | [optional] |
| Maximum retries per query | [value] | [optional] |
| Maximum conversation turns | [value] | [optional] |

## 3. Agent Scaffolding and Environment

**Memory/context settings:** [Describe conversation memory settings, context window usage, state management between trials]

**Tools/plugins available:** [List all tools, plugins, or function calls available to the agent; N/A if none]

**Execution environment:** [Describe sandbox, code interpreter settings, or external systems accessible]

**Multi-agent configuration:** [If applicable: number of agents, roles, interaction topology, communication protocol]

## 4. Prompt Specifications

**System message/instructions:** [Provide complete text of any system-level instructions]

**User instructions:** [Insert complete prompt text here, exactly as presented to the model. Include all formatting, line breaks, and special characters.]

**Few-shot examples (if applicable):** [Specify number and list examples; describe how examples were selected]

**Retrieval-Augmented Generation (if applicable):**

**Retrieval mechanism:** [Describe retrieval system; N/A if not using RAG]

**Document corpus:** [Describe corpus source and size]

**Retrieval parameters:** [Number of documents retrieved, similarity threshold, etc.]

**Filtering/ranking procedures:** [Describe any post-retrieval processing]

## 5. Pilot Testing and Development History

**Number of prompt iterations tested:** [Approximate count of prompt variants explored before finalizing]

**Models tested during development:** [List all models tried during piloting, not just the final selection]

**Parameter configurations explored:** [Describe range of temperatures, sampling parameters, etc. tested]

**Did preliminary results inform hypotheses?** [Yes/No. If yes, describe how.]

**Selection rationale:** [Explain why the final model/prompt/parameters were selected]

**Pilot data retention:** [Will pilot data be retained and made available? Where?]

# 6. Hypotheses

*State each hypothesis precisely. Specify directionality where applicable.*

**Primary hypotheses:** *[State all primary hypotheses]*

**Secondary hypotheses:** *[List any secondary or exploratory hypotheses]*

# 7. Experimental Design

**Design type:** *[Between-subjects / Within-subjects / Mixed / Other (specify)]*

**Independent variable(s):** *[List IVs and their levels]*

**Dependent variable(s):** *[List DVs and how they will be measured]*

**Control conditions:** *[Describe any control or baseline conditions]*

**Number of trials/completions per condition:** *[Specify]*

**Total number of API calls planned:** *[Specify]*

**Power analysis or sample size justification:** *[Describe rationale for sample size]*

**Stopping rule:** *[When will data collection end? Fixed n / sequential / other]*

# 8. Response Processing

**Response format expected:** *[Free text / JSON / Multiple choice / Likert scale / etc.]*

**Parsing procedure:** *[Describe how responses will be extracted from model output]*

**Handling ambiguous responses:** *[What happens if response doesn't match expected format?]*

**Non-response/timeout handling:** *[How will incomplete responses be handled?]*

**Response-level exclusions:** *[Criteria for excluding individual responses]*

**Automatic retry conditions:** *[Under what conditions will queries be retried?]*

**Maximum retries:** *[Number]*

**Retry parameter changes:** *[Will any parameters change on retry? If so, which?]*

# 9. Analysis Plan

**Primary outcome measure:** *[Precise definition of main DV]*

**Measurement/coding procedure:** *[How will raw responses become analyzable data?]*

**Secondary outcome measures:** *[List any additional outcomes]*

*For each hypothesis, specify the exact statistical test and decision rule:*

| Hypothesis | Statistical Test | Decision Rule | Effect Size |
|---|---|---|---|
| H1 | *[test]* | *[e.g., p < .05]* | *[measure]* |
| H2 | *[test]* | *[rule]* | *[measure]* |

**Planned exploratory analyses:** *[Describe any analyses not tied to specific hypotheses]*

**Correction method:** *[Bonferroni / FDR / None / Other (specify)]*

## 10. Robustness and Generalization Analyses

**Additional models to test:** *[List all models]*

**Prompt variants to test:** *[List all variants]*

**Variation dimensions:** *[Wording / Format / Instruction ordering / etc.]*

**Parameters to vary:** *[List parameters (and ranges) for sensitivity analysis]*

**Planned specification curve:** *[If conducting multiverse analysis, describe the space of specifications]*

## 11. LLM-as-Judge Specifications (if applicable)

*Complete this section if using an LLM to evaluate or score responses.*

**Evaluator model:** *[Specify model and version used for evaluation]*

**Evaluation prompt:** *[Provide complete evaluation prompt verbatim]*

**Rating scale/criteria:** *[Describe the rating scale or evaluation criteria]*

**Evaluator refusal handling:** *[What if the evaluator refuses to rate?]*

## 12. Staged/Adaptive Design Rules (if applicable)

**Stage structure:** *[Describe stages of the study]*

**Decision rules between stages:** *[Specify exact criteria for proceeding to next stage]*

**Example decision rules:**
- *If pilot accuracy < 20%, increase context length by X tokens*
- *If ceiling effects (accuracy > 90%), switch to harder item set B*
- *If refusal rate > 30%, add instruction clarification C*

**Your decision rules:** *[Specify your adaptive rules here]*

## 13. Transparency and Data Sharing

**Materials to be shared (check all that apply):**
- ☐ Complete prompt text and all variants
- ☐ Raw model outputs (full API responses)
- ☐ Processed/coded data
- ☐ Analysis code
- ☐ Pilot data and development logs
- ☐ Model API logs with timestamps

**Repository location:** *[Where will materials be deposited? e.g., OSF, GitHub, Zenodo]*

**Data availability timeline:** *[When will data be made available?]*

**Access restrictions:** *[Any restrictions on data access? If so, justify]*

## 14. Attestation

**By submitting this preregistration, I/we attest that:**

1. Data collection for the confirmatory analyses described herein has not yet begun.
2. The hypotheses, methods, and analysis plans described represent our genuine intentions prior to data collection.
3. Any deviations from this preregistration will be transparently disclosed and justified in resulting publications.
4. All pilot testing and development history has been accurately described.
5. Results from all pre-specified robustness analyses will be reported regardless of outcome.

**Signature(s):** _______________________________________________

**Date:** _________________________________________________

*Template adapted from: Position: AI Safety Evaluations Should Focus on Human-AI Systems (ICML 2026)*