# OpenReview forum: "Position: Preregister Experiments with AI Agents"
_ICML.cc/2026/Position_Paper_Track — ICML 2026 Position Paper Track spotlight_

### Official Review · Reviewer_Vpex · 2026-02-23

**Significance:** 4
**Argument Clarity:** 4
**Rating:** 6
**Confidence:** 4

**Questions:**

I'm curious why you limited the position to be behavioral experiments with AI agents and not all ML experiments. I can guess at the answer from Figure 2, but I would argue that traditional ML experiments still have significant specification flexibility (even just through the set of hyperparameters). Figure 2 also suggests that specification flexibility is much higher for human subject experiments than traditional ML experiments - I'm curious why you think this is the case?

Overall it seems to me that the argument largely applies to traditional ML experiments as well. I could see the position being expanded to incorporate all ML experiments, not just those with AI agents as participants.

**Alternative Views Section:**

Yes

**Compliance With Llm Reviewing Policy A Conservative:**

Affirmed.

**Discussion Potential:**

4

**Final Justification:**

For the most part, see "Strengths". The authors provided some additional valuable discussion in the rebuttal.

**Paper Summary:**

The paper proposes that the field of machine learning adopt a convention of pre-registering experiments with AI agents, similar to the existing pre-registration convention in the social sciences. "Pre-registration" means publicly committing in advance to the exact experiment design, including which models to use, the prompts, the hyperparameters, the analysis that will be done, etc. An experiment which is not pre-registered leaves the door open for researchers to adjust the experiment design after seeing initial results, which reduces the validity of the experiment (even if the researchers are not intentionally p-hacking). The paper argues that the replication issues that caused the social sciences to adopt this convention not only apply here, but are in fact exacerbated by the usage of AI agents in place of human participants. Based on my reading, the main categories of reasons are 1) experiments with AI agents are lower cost so it is easier to rerun them after seeing initial results and 2) researchers have control over not just the experiment design but the agents themselves, and more researcher degrees of freedom means more opportunities to cherry-pick results.

**Position:**

Yes

**Position In Title:**

Yes

**Related Work:**

3

**Strengths And Weaknesses:**

**Strengths**

The authors' argument is clearly reasoned and supported well. Each point of the argument is either supported with citations or by carefully justified argumentation. In addition to the direct rhetorical argument, the paper presents the argument in several other ways, including through visual aids. I think this varied approach will make this paper accessible to a greater number of readers. In addition to arguing why pre-registration should become the norm, the paper also discusses what properties a pre-registration template should have and provides an initial draft of such a template. This makes the paper not just an general opinion piece, but a concrete proposal.

**Weaknesses**

While I think the Alternative Views section is acceptable as-is, I think the paper does not fully grapple with the obstacles to such a proposal. The proposal would constitute a major change in how ML research is conducted and I think this type of step change would face significant opposition, regardless of the rational merits of the argument. I think the paper would benefit from a discussion of a practical path to the desired end goal of full pre-registration, rather than just the end goal. For example, I would view NeurIPS-style reproducibility checklists as a step on this path. Or if the authors believe that a step change is practical, some discussion of that would be welcome too.

I also think that one of the biggest obstacles is not really discussed: incentives. Researchers are often under pressure to publish, which creates the incentive to cherry-pick publishable results. Researchers have an incentive to resist a proposal which may make it harder for them to publish. While this is clearly not a logically valid counterargument, I expect this to be a major obstacle in practice. If the paper is serious about trying to make pre-registration the norm, I think a more thorough discussion of this obstacle is important.

**Minor comments**

1. I think this statement is excellent: "The goal is not to eliminate flexibility, but to make flexibility visible." Consider spotlighting this earlier in the paper somehow.
2. I didn't really understand Alternative View 1. Is Horton arguing because of low marginal cost, you could pre-register your experiment and then if you don't get what you want, just pre-register and run another version?
3. I think the paper sometimes overstates how low the marginal cost of ML experiments is. For example, Figure 2 puts the marginal cost per run of both "Experiments with AI Agents" and "Traditional ML Evaluations" close to "seconds / pennies / API call". This seems like an exaggeration, especially for any experiments involving LLMs. My impression is that LLM papers commonly use hundreds to thousands of GPU-hours. Maybe the authors meant marginal cost per training / test example? If not, I'm curious what types of experiments they have in mind that take seconds / pennies / one API call. However, I agree that the cost remains lower than experiments with human participants, so I don't think this affects the core argument.

**Support:**

3

---

> ### Author Rebuttal · Authors · 2026-03-26
>
> Thank you for your thoughtful evaluation and assessment that our argument is clearly reasoned and well supported, that the varied presentation approach (visual aids, taxonomy, template) makes the paper accessible to a broad audience, and that the inclusion of a concrete template elevates the paper beyond a general opinion piece. Your points about addressing the practical obstacles to adoption and expanding our argument to other ML experiments have also further strengthened our paper. We address each one below:
>
> **Path to adoption**
>
> We agree that preregistration would constitute a major change in how ML research is conducted, and we appreciate your point that its rational merits alone do not guarantee adoption. Drawing on the example of preregistration in the social sciences, we argue that preregistration in the ML community should begin with voluntary early adopters, followed by institutional incentives, and let these steps lead to normalization. In the social sciences, preregistration began with a small number of researchers voluntarily using the Open Science Framework (OSF), followed by journals introducing Registered Reports and open-practice badges, followed by funders incorporating preregistration into grant requirements, and is now considered standard practice for confirmatory research. We outline a parallel pathway for experiments with AI agents, which we organize by stakeholder.
>
> For researchers, the natural first step is voluntary preregistration on existing platforms. OSF already supports custom templates, and our proposed template (Supplementary Materials) can be used immediately—-no new infrastructure is required.
>
> For conferences and journals, we propose a graduated approach rather than immediate mandates. As a first step, venues could add a question to their submission checklist: “Was this study preregistered? If yes, provide the registration link. If no, briefly explain why.” This imposes minimal burden while making the norm visible and normalizing the practice of considering preregistration. A second step would be introducing open-practice badges for preregistered studies, following the model adopted by Psychological Science and over 80 other journals, which were followed by an increased adoption of open practices. A third step—requiring preregistration or offering a Registered Reports track—could follow once the community has sufficient experience with the practice. As you note that NeurIPS already requires a reproducibility checklist, and ICML has adopted similar practices, and extending these checklists to include preregistration-related questions for studies involving AI agent behavioral experiments would be another natural and incremental step.
>
> For funding agencies, we recommend that calls for proposals involving experiments with AI agents include a preregistration plan as a standard component of the research design section.
>
> We will incorporate these points in Section 5.
>
> **Expanding argument to all ML experiments**
>
> We agree that many of the concerns we raise—specification search, selective reporting, undisclosed flexibility—are relevant to ML research more broadly. We focus specifically on experiments with AI agents because they occupy a uniquely vulnerable position: traditional ML experiments have developed meaningful safeguards over the past decade (standardized benchmarks, held-out test sets, reproducibility checklists, model cards), whereas behavioral experiments with AI agents currently lack analogous protections—there is no "test set" to hold out when the experiment is the evaluation, no standardized benchmark when prompts are bespoke research instruments, and no established reporting norms for the degrees of freedom we catalog in Table 1. That said, we share your intuition that the argument extends further, and we will include a discussion at the end of the revised paper addressing the broader applicability of preregistration to ML experiments more generally.
>
> **Marginal cost of ML experiments**
>
> We appreciate this clarification and agree the figure should be more precise. The research paradigm we address does not involve training or fine-tuning LLMs—it involves prompting existing models via API to elicit responses to structured behavioral tasks (economic games, moral dilemmas, survey questions, etc.). In this setting, the marginal cost of an additional experimental run genuinely can be seconds and pennies: a single API call to GPT-4 or Claude costs fractions of a cent, and our illustrative anchoring simulation ran thousands of simulations at a relatively small expense (<$100) (see response to Reviewer 6iSg for more details). This is quite different from GPU-intensive training and evaluation pipelines, which we agree are substantially more expensive. We have revised the paper to clarify that our cost characterization applies specifically to inference-time behavioral experiments with API-accessed models, not to ML research involving training or large-scale compute.

---

> > ### Author Rebuttal · Reviewer_Vpex · 2026-04-03
> >
> > Thanks for the thoughtful response and I appreciate the discussion. The "Path to adoption" and "Expanding argument to all ML experiments" responses make sense to me. However, I remain unconvinced about the marginal cost point. Even for experiments with only inference and no training, I think "pennies / seconds" remains a significant exaggeration. In my experience, \\$50-200 for API costs is a typical amount for a full set of experiments, which aligns with your <\\$100 number. To me, \\$50-200 is much more than "pennies". For open-weight models, in my experience a single inference pass typically takes on the order of a second, so we're still looking at tens to hundreds of GPU hours if you're using a sizable dataset like MMLU and especially if you're using multiple models in multiple experimental configurations, etc.
> >
> > To reiterate, I don't think the difference between "pennies" and "$50-200" affects the validity of the paper's argument. I just think it's not very accurate, having run many LLM experiments myself (both open-weight and via proprietary API).
> >
> > However, I don't insist on this change and I will maintain my Strong Accept score.

---

### Official Review · Reviewer_DUQ8 · 2026-03-10

**Significance:** 3
**Argument Clarity:** 3
**Rating:** 4
**Confidence:** 4

**Questions:**

Related to the second point of the weaknesses: What is the limitation of the current practice in ML that releases the full (or at least a significant portion of) codebase to reproduce the main experiments, and how does pregistration remedy the limitation?

**Alternative Views Section:**

Yes

**Compliance With Llm Reviewing Policy A Conservative:**

Affirmed.

**Discussion Potential:**

3

**Final Justification:**

The authors' comment addressed my concerns in detail, and I am happy to see this paper at the conference.

**Paper Summary:**

This paper suggests preregistering AI agent experiments in a public registry to foster the credibility and reproducibility of experiments. The paper first discusses experiment preregistration as a well-established standard in human subject research and appeals to its necessity and usefulness in AI agent research. By preregistering the experimental details with the proposed protocol, the authors argue that transparency and reproducibility of agent-based experiments can be improved.

**Position:**

Yes

**Position In Title:**

Yes

**Related Work:**

3

**Strengths And Weaknesses:**

## Strengths

* Strong position, clearly stated and well supported.
* Preregistration is already an established practice in other fields, and therefore, its efficacy is tested to a certain extent.
* Preregistration is a practice that is novel to the machine learning community, and the discussion around it may be beneficial for the ML community.


## Weaknesses

* The paper could greatly benefit from providing more examples. A few of them are mentioned in the introduction, but it is still unclear what kind of experiments are eligible for preregistration. There are many kinds of experiments that can be done with AI agents, and it's unclear whether the paper is proposing that all kinds of such experiments will benefit from preregistration.
* The current mainstream practice in the ML community for fostering reproducibility is simply open-sourcing the entire project. The community can then evaluate and validate the pipeline as much as needed. This practice has been feasible because the cost is (relatively) low and reproduction is (relatively) easy. This point may be related to Alternative View 1, but requires deeper discussion, as this is the standard practice in ML nowadays.

**Support:**

3

---

> ### Author Rebuttal · Authors · 2026-03-26
>
> We appreciate the recognition that the position is clearly stated and well supported, that preregistration's strong track record in other fields strengthens the case for adoption, and that introducing this discussion to the ML community is itself a valuable contribution. The other points you raised have led to substantive improvements in the revised manuscript. We address each one below:
>
> **More examples**
>
> The range of research designs that fall under "experiments with AI agents" is broad as they share a structural vulnerability. Whether a study tests how LLMs respond to moral dilemmas, framing effects, cultural differences, or moderation decisions, the same issue arises: many reasonable design choices can affect the result. One outcome is often more publishable than the other. The low marginal cost of running additional configurations also means the full space can be traversed relatively quickly and cheaply. Any study that draws an inferential conclusion from this kind of pipeline is a candidate for preregistration. Our new example on anchoring (see response to Reviewer 6iSg) provides a concrete example of why: across over 2,000 defensible configurations, the anchoring effect can range from null to strongly positive, so a researcher can arrive at different conclusions depending on which path, or even set of paths, through the specification space they report.
> That said, not all research involving AI agents should require preregistration. Purely exploratory work—probing a new model's capabilities, iterating on prompts during development, generating hypotheses from unexpected behaviors—is valuable and should remain unconstrained, provided it’s properly communicated as such.
>
> **Reproducibility vs. credibility**
>
> We agree that open-sourcing code and data solves the problem of reproducibility: can someone else run the same pipeline and get the same numbers? For standard ML evaluation—training a model, running it on a benchmark, reporting accuracy—this is often sufficient, because the key claims are about performance on a fixed task with a fixed metric, and the open-sourced pipeline fully specifies the path from data to result. We have no quarrel with this practice and strongly endorse it. Our template, in fact, includes open-sourcing as a core transparency commitment.
>
> But behavioral experiments with AI agents face a different problem that open-sourcing alone cannot solve: the problem of undisclosed specification search. When a researcher open-sources the code for their final reported experiment—the one prompt, model, and setting that produced the published results—they have made that single path fully reproducible. What they have not revealed is the garden of forking paths that preceded it: the prompt variants they tried before settling on the one that worked, the other models that produced null results, the temperature settings they explored, or the parsing rules they adjusted after seeing initial outputs. This is the distinction between reproducibility (can I get the same result from the same pipeline?) and credibility (should I believe this result reflects a robust phenomenon rather than a specification that was optimized to produce it?). We will add a discussion of this distinction with our illustrative simulation on anchoring (see response to Reviewer 6iSg for more details).
>
> Importantly, the open-sourcing approach also places the burden of detecting problems on the wrong people. When the community's credibility mechanism is "release the code and let others validate it," the work of identifying specification search, testing robustness to alternative configurations, and assessing whether the reported path was cherry-picked falls entirely on reviewers, readers, and replicators. In practice, this means the burden may not get discharged at all: reviewers operating under tight deadlines may not rerun an experiment across alternative specifications, and independent researchers may not conduct systematic robustness checks of others’ work, especially if they are under tight research budgets and the cost of API calls to LLMs begins to compound. At this point, the work is also already published and potentially informing many high stakes decisions. The result is a system where reproducibility is theoretically possible but credibility is not sufficiently verified. Preregistration inverts this burden. It asks the original researcher, the person with full knowledge of the experimental history and the lowest marginal cost of documentation, to make their specification choices transparent.
>
> That said, we want to note that the two practices are complementary, not competing. The strongest evidence comes from studies that are both preregistered and open-sourced: the preregistration establishes what was planned, the open-sourced code establishes what was executed, and any discrepancies between the two are visible and must be justified. We have revised Sec. 5 to make this distinction and complementarity explicit.

---

> > ### Author Rebuttal · Reviewer_DUQ8 · 2026-04-04
> >
> > I appreciate the response by the authors.

---

### Official Review · Reviewer_h8BJ · 2026-03-13

**Significance:** 4
**Argument Clarity:** 4
**Rating:** 5
**Confidence:** 4

**Questions:**

Do you have mpirical evidence of how prevalent the problems you're mentioning actually are in the current literature? This might be difficult to attain, but it would really strengthen your argument

**Alternative Views Section:**

Yes

**Compliance With Llm Reviewing Policy A Conservative:**

Affirmed.

**Discussion Potential:**

3

**Paper Summary:**

The paper comments on issues with the new emerging paradigm where LLMs or AI agents are often used as participants in studies of cognition, decision-making, and social dynamics. Human subjects research has previously experienced replicability crises due to the huge numbers of degrees of freedom that researchers have in experimental design; the paper argues that the cheapness and accessibility of AI tools amplify the methodological vulnerabilities that led to this crisis. Social experiments with AI agents also have high specification flexibility (prompt wording, model selection, decoding parameters, parsing rules, etc.), but also have near-zero marginal cost per run. Thus, researchers can explore numerous experimental configurations until whatever result they want emerges, often without any mention of this in their results.

The paper outlines the researcher degrees of freedom specific to AI agent experiments, also providing a taxonomy. The authors then argue that existing ML reproducibility norms (held-out test sets, standardized benchmarks) do not address the problems introduced by these degrees of freedom, because behavioral experiments have no "test set" to hold out and no standardized protocol for prompts.

The concrete proposal is that preregistration (publicly committing to hypotheses, methods, and analysis plans before data collection) should become standard practice for behavioral-style research that uses AI agents. To this end, the authors propose a preregistration template tailored to AI experiments.

**Position:**

Yes

**Position In Title:**

Yes

**Related Work:**

3

**Strengths And Weaknesses:**

Strengths
* The problem the paper addresses is important and timely: the lack of methodological safeguards in behavioral experiments with LLMs. The authors make a compelling case that the combination of high experimental specification flexibility and low marginal cost of experiments with AI agents creates an environment where research results can easily be hacked or completely accidental
* The taxonomy of researcher degrees of freedom (table 1) is very useful. It makes concrete what would otherwise be a vague concern about "flexibility," which makes the paper much more approachable to new readers
* The paper is very well written: well-structured, clearly argued and following a logical flow

Weaknesses
* The paper coudl be framed more appropriately for an ICML audience. When ML researchers hear "experiments with AI agents," behavioral experiments are not the first thing that comes to mind; the paper could do a better job of establishing how the research field they're concerned with differs from the one the audience is likely coming from. Paragraph 3 of Section 1 does briefly explain how its concerns are different from more standard ML reproducibility issues, but I think this distinction deserves much more attention and  more prominent treatment.
*The paper focuses almost exclusively on behavioral experiments but the introduction also mentions AI agents operating autonomously in high-stakes domains (negotiating contracts, managing portfolios). These are quite different contexts with different concerns about methodolgy, but the paper doesn't clearly address whether the proposed template applies equally to both.
* The paper could do more to address the practical question of enforcement and adoption

**Support:**

4

---

> ### Author Rebuttal · Authors · 2026-03-26
>
> Reviewer h8BJ
> Thank you for your very helpful review. We are grateful for the recognition that the problem we address is important and timely, that our taxonomy of researcher degrees of freedom (Tab. 1) makes our concern concrete and understandable to readers, and that our paper is well-structured and clearly argued. We also appreciate your suggestions about framing for an ICML audience, clarifying types of experiments with AI agents, and addressing enforcement and adoption—each of which has led to meaningful improvements in the revised manuscript. We address each point below:
>
> **Framing for ICML**
>
> We recognize that “experiments with AI agents” may initially evoke benchmark evaluations, leaderboard comparisons, or reinforcement learning experiments for some ICML readers. The research paradigm we address is different: it involves using LLMs as participants in structured behavioral tasks where the goal is to characterize the agent's reasoning, biases, or decision-making rather than to optimize a performance metric against a held-out test set. This distinction matters methodologically because the standard ML safeguards against overfitting (fixed train/validation/test splits, standardized benchmarks, held-out evaluation sets) do not cleanly apply when the experiment is the evaluation and prompts are customizable research instruments rather than standardized inputs. We have expanded Sec. 1 to make this distinction more prominent and explicit.
>
> **Types of experiments with AI agents**
>
> We agree that our paper references two related contexts: (a) behavioral experiments that use AI agents as proxies for human participants, and (b) the evaluation of AI agents operating autonomously in high-stakes (or low-stakes) domains. We argue that (a) and (b) are more similar than they initially appear, because they share the same fundamental methodological vulnerability. In both cases, researchers face a large specification space—which model, which prompt scaffolding, which scenario, which success metric, for just a few examples—and the same incentive to report favorable configurations. A researcher testing whether LLM-simulated consumers respond to scarcity cues navigates the same garden of forking paths as a researcher testing whether a hiring agent screens resumes without gender or racial bias: both can vary the framing, the model, the parameters, and the evaluation criteria until a publishable result emerges. We will revise Sec. 1 and Sec. 4 to clarify that the template is designed to cover both contexts.
>
> **Enforcement and adoption**
>
> See response to Reviewer Vpex (Path to adoption) for how we plan to address the practical question of adoption.
>
> We want to be transparent about the limits of enforcement. Even existing preregistration systems for human-subjects research cannot prevent a determined bad actor from running experiments secretly and then registering select ones post hoc. But enforcement does not require perfection to be effective—it requires shifting defaults and raising costs. As we discuss in Sec. 6 (Alternative View 2), preregistration creates an auditable record that makes deception riskier, especially when combined with norms of sharing raw API logs and analysis code. The goal is not to police every study but to change the culture of how experiments with AI agents are conducted and reported—making transparency the path of least resistance rather than an afterthought. See response to Reviewer 6iSg (Manipulation) for more details.
>
> **Empirical evidence**
>
> See response to Reviewer 6iSg (Real examples) for details about a new, concrete example of the problem of the garden of forking paths in experiments with AI agents.
>
> We can also draw on analogy from adjacent fields where prevalence has been measured. John et al. (2012) surveyed psychologists and found that a majority admitted to at least one questionable research practice, including selectively reporting studies that "worked." We find it plausible that ML researchers can be similarly susceptible to these pressures. We are not aware of an equivalent survey of researchers conducting experiments with AI agents, though we would welcome such work. In the meantime, our anchoring demonstration makes the potential for specification-driven variability tangible: we generated over 2,000 defensible specifications whose results range from strong anchoring to no effect, illustrating just how easy it would be for a researcher to—consciously or not—select a flattering subset to report.

---

> > ### Author Rebuttal · Reviewer_h8BJ · 2026-04-03
> >
> > The authors show willingness to adjust aspects of the paper that will resolve my concerns

---

### Official Review · Reviewer_6iSg · 2026-03-20

**Significance:** 2
**Argument Clarity:** 2
**Rating:** 4
**Confidence:** 3

**Questions:**

See weakness!

**Alternative Views Section:**

Yes

**Compliance With Llm Reviewing Policy A Conservative:**

Affirmed.

**Discussion Potential:**

3

**Paper Summary:**

This position paper argues for preregistration practices in experiments involving AI agents. The authors identify a key methodological risk， i.e.，the large space of researcher degrees of freedom, which enables implicit specification search and undermines the credibility of findings. The paper proposes a structured preregistration template tailored to AI-agent-based experiments and provides recommendations for researchers.

**Position:**

Yes

**Position In Title:**

Yes

**Related Work:**

2

**Strengths And Weaknesses:**

The paper targets a rapidly growing paradigm, i.e., LLM-based behavioral experiments, and the proposed preregistration template is concrete and actionable.

However, the paper is entirely conceptual / normative, which doesn't cover any real examples of inflated results or empirical quantification of specification search.  The core idea is not new as well.

While the authors acknowledge that preregistration could potentially be manipulated,  researchers can easily complete all experiments in secret and only then submit a preregistration that mirrors those results. The paper lacks a robust technical framework to verify that data collection truly commenced only after registration.

**Support:**

2

---

> ### Author Rebuttal · Authors · 2026-03-27
>
> Thank you for reviewing our paper and recognizing its contribution to a rapidly growing paradigm with a concrete, actionable template. We are also grateful for your constructive feedback, which has substantially strengthened our paper. We address each point below:
>
> **Real examples**
>
> By its nature, undisclosed specification search is difficult to identify in published work—if the flexibility were visible, the problem would be less concerning. But indirect examples suggests inflated results exist: preregistered studies in psychology consistently report smaller effect sizes and more null results than non-preregistered studies (Schäfer & Schwarz, 2019; Scheel et al., 2021), and prompt sensitivity studies demonstrate that small wording changes produce large swings in LLM behavior (Sclar et al., 2024). To provide a more concrete example, we conducted a new simulation examining the anchoring effect in LLMs across 2,430 experimental conditions. The simulation demonstrates how a researcher can arrive at dramatically different conclusions depending on which path they report through the specification space. They could even report something like: “Across three models from different families and architectures, we find consistent evidence that LLMs exhibit human-like anchoring biases,” a claim that sounds like a nice robustness check. The robustness check is real in a narrow sense—three models were tested—but it was selected from a much larger space of possible robustness analyses, most of which would have told a different story. We will unpack these examples in the additional page of the camera-ready version. Full code, data, and figures are also available at https://osf.io/5rc2h/overview?view_only=53e0a1748b4943e4989a2203a1ba9c81.
>
> **Novelty**
>
> We acknowledge that preregistration itself is a well-established practice—that is, in fact, central to our argument. Our contribution is not its invention but its extension to a field that lacks analogous safeguards. Specifically, we provide: 1) a comprehensive taxonomy of researcher degrees of freedom specific to experiments with AI agents (Table 1), 2) an analysis of why these experiments are uniquely vulnerable, they combine high specification flexibility with low marginal cost (Figure 2), and 3) a preregistration template tailored to experiments with AI agents that addresses their novel degrees of freedom, which don’t exist in current preregistration frameworks. Also, note that when PROSPERO introduced systematic review registration, preregistration was already decades old in clinical trials, yet it was recognized as a meaningful contribution because meta-analyses required field-specific adaptation. We make the same case for experiments with AI agents. We will highlight these points in our revised Sec. 1.
>
> **Manipulation**
>
> We agree, and we expand upon this point in Sec. 6 (Alternative View 2). Preregistration is not a guarantee of honesty—a researcher could explore widely and then preregister only the final configuration that worked. But without preregistration, this specification search is invisible and carries less reputational risk; with it, a dishonest researcher must lie, a materially different act that carries more meaningful consequences. Importantly, the relevant comparison is not also preregistration vs. a perfect system but preregistration vs. the current status quo, which has essentially no safeguards.
> Also, the same manipulation concerns also apply to preregistration in clinical trials and psychology. OSF and AsPredicted cannot verify that a researcher did not collect or examine their data before registering their hypotheses and analysis plans. ClinicalTrials.gov cannot verify that no patient data were collected before a trial was registered; PROSPERO cannot confirm that a systematic review team had not already screened articles before submitting their protocol. Yet the empirical evidence suggests it helps: preregistered studies report smaller effect sizes and more null results than non-preregistered counterparts (Schäfer & Schwarz, 2019; Scheel et al., 2021; van den Akker et al., 2024). Ultimately, we view preregistration the way we view locks on doors: it does not stop a determined burglar, but it does deter accidental or opportunistic entry, raise the cost of violation, and establish a norm that makes transgression recognizable. The field is currently operating without locks.

---

> > ### Author Rebuttal · Reviewer_6iSg · 2026-04-05
> >
> > Thank the authors for the response.

---

### Decision · Program_Chairs · 2026-04-30

**Decision:**

Accept (spotlight)

**Comment:**

This paper clearly presents a very timely and important position promoting the pre-registration of behavioural experiments with AI agents. As this is not common practice within the machine learning community, presentation of the paper could spark important interdisciplinary discussions and awareness of best practice that is directly applicable to a wide range of current research within the community.